# Fatigue Life of RC Bridge Decks Affected by Non-Uniformly Dispersed Stagnant Water

**Eissa Fathalla** [1] **, Yasushi Tanaka** [2] **and Koichi Maekawa** [3,*]

1    Department of Civil Engineering, The University of Tokyo, 7-3-1 Hongo, Bunkyo-ku, Tokyo 113-8656, Japan; eissa.tokyo.concrete@gmail.com
2    Department of Civil and Environmental Engineering, Kanazawa Institute of Technology, 7-1 Ohgigaoka, Nonoichi, Ishikawa 921-8501, Japan; ytanaka@neptune.kanazawa-it.ac.jp
3    Department of Civil Engineering, Graduate School of Urban Innovation, Yokohama National University, 79-1 Tokiwadai, Hodogaya, Yokohama 240-8501, Japan
*    Correspondence: maekawa-koichi-tn@ynu.ac.jp; Tel.: +81-45-339-4155



**Featured Application: This study aims to achieve the deterioration rank of the site-inspected wetting locations of road bridge decks, where the wetting locations can be detected by non-destructive testing technology. Thus, rational life assessment of bridge decks is secured.**

**Abstract:** Stagnant water on reinforced concrete (RC) decks reduces their life significantly compared to the case of dry states. Fully submerged states have been investigated as the most severe case, which is however rarely experienced in reality. Currently, it is possible to simulate concrete–water interactions for lifetime prediction of RC decks. In this study, fatigue lifetime is systematically computed for various locations of stagnant water at the upper layer of RC decks. It is found that the patterns of wet and dry areas have a great influence on the remaining fatigue life even though the same magnitude of cracking develops. Then, a hazard map for the wetting locations with regard to the remaining fatigue life is presented based on the systematically arranged simulation. Finally, a nonlinear correlation is introduced for fatigue life prediction based upon site inspected wetting locations, which can be detected by non-destructive testing technology.

**Keywords:** multi-scale simulation; fatigue loading; road bridge decks; stagnant water

## 1. Introduction

Reinforced concrete (RC) bridge decks suffer from high deteriorations due to environmental attacks besides traffic loading, where corrosion, freeze and thawing, alkali silica reaction, and shrinkage and thermal cracking were reported to be significant on the reduction of life of RC decks [1–9].

On the other hand, in high seismic risk countries like Japan, thicknesses of RC decks were aimed to be thinner in order to reduce the inertia forces at earthquakes to satisfy earthquake resistant design requirements since bridge slabs are main source of seismic loads to bridges. These limited-thicknesses of decks (less than 200 mm) were constructed in 1960–70s, where enormous numbers of highway bridges were built. After around half a century, degradation of bridge decks has been observed from accumulated loads of daily traffics, where these deteriorations may reduce the safety of users.

Previous research reported that RC slabs exposed to moving loads are extremely deteriorated compared to those exposed to fixed-point pulsating ones. The reversal cyclic-shear along crack planes induced by moving wheel type loading rapidly deteriorates the shear transfer of aggregates interlock along concrete cracks [10]. Finally, RC slabs speedily lose their stiffness until total failure. Thus, we have common issues in view of bridge deterioration where mechanical fatigue loads and environmental actions develop together with more or less interaction.

It should be noted that the performance of RC decks can be upgraded by utilizing pre-stressing techniques [11] for newly constructed bridges. In fact, traffic-induced cracking is suppressed and water may not come inside crack gaps of concrete. However, we have to face serious problems such as that damaged RC decks of many bridges cannot be easily replaced since it will directly disturb the traffic flow leading to social problems. Therefore, old RC decks shall be retrofitted and/or limitedly replaced for extending the lifetime of RC decks on the basis of reliable maintenance plans.

Moreover, stagnant water from rainfall may remain on RC bridge decks due to imperfect waterproofing works and/or insufficient maintenance. Stagnant water is well recognized to seriously influence fatigue life [12–17], where performance of concrete is weakened and disintegration between aggregates and cement binder has been reported from experiments and site investigations, as shown in Figure 1. Accelerated wearing of the surface of cracks is demonstrated experimentally by cyclic shear tests in pure water, and the loss of the strength of concrete is qualitatively explained by changing the surface energy of Calcium-Silicate-Hydrate (CSH) binders. Under a high deformational rate, condensed water cannot easily disperse leading to a sharp rise and/or fall of the pore water pressure owing to its viscosity [18]. Fatigue loading experiments of water supply on RC decks demonstrate the negative impacts on their lives, and it was reported that the reduction in fatigue life is around 1/200 of the dry states [12,13].

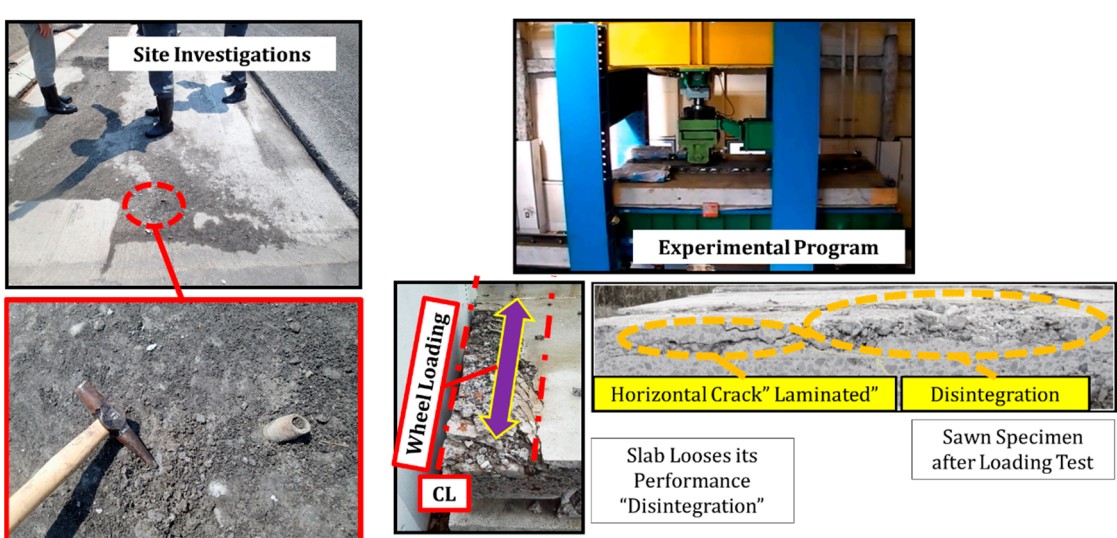

**Figure 1.** Site and experimental investigations for the disintegration phenomenon.

In recent research [17], predictive models were proposed for fatigue life of RC decks based on site inspected cracks under stagnant water, where artificial neural network model and mechanics-based correlation are introduced on the basis of an enormous number of investigated crack patterns. However, in the mentioned research, the RC decks are assumed to be in fully submerged states for safer and conservative assessment of RC decks at the current moment. It was found that cracks with the existence of stagnant water have more negative impacts on fatigue life of RC decks than the cracks in dry states, but it should be noted that the negative impacts of stagnant water on fatigue life are much higher than the one of the cracks as an overall.

Most of these research usually has targeted RC decks in fully submerged conditions, but little attention has been directed to more realistic moisture states where the wet area is scattered in space. Nowadays, some multi-scale simulation can deal with the crack to water interaction on the basis of upgraded constitutive laws for fatigue and Biot's theory [19–21], where spatially non-uniform wetting and dry regions can be numerically reproduced. On the other hand, in the past few decades, there were no quick-feasible technologies for detecting the wetting regions in real RC decks on site. However, currently, Mizutani et al. [22] are developing a fast detection technique for wetting locations in the

upper layer by signal processing of UHF-band ground penetrating radar. This technique has been validated at the site and partially utilized in bridge maintenance in Japan.

In this study, the authors investigate the remaining fatigue life of various wetting regions in the upper layer of RC decks by utilizing the multi-scale simulation. It is empirically known that locations of stagnant water substantially affect the remaining fatigue life and that the wheel-loading path is the highest risk area that may reduce the fatigue life. Then, a detailed hazard map on wetting locations in between pavement and the concrete decks is aimed to meet the challenge of rational life assessment. Finally, a predictive correlation is proposed for fatigue life of RC decks based on site inspected wetting locations with high accuracy.

## 2. Methodology for Predicting the Remaining Fatigue Life

Figure 2 shows the methodology for predicting the remaining fatigue life of RC decks based upon site inspected wetting locations on the surface of decks as follows:

1.  Ground penetrating radar is installed on a vehicle that runs about 80 km/h over the bridge, where signal responses of the hidden information of RC decks below the pavement layers are detected [22].
2.  These signals are processed to achieve sound locations of the water especially at the upper surface layer of RC decks.
3.  These wetting locations are induced into the finite element model by utilizing the multi-scale simulation program.
4.  Travelling wheel load is applied until the failure of RC decks based on a failure criterion that will be stated in a later section.
5.  Finally, remaining fatigue life of RC decks is computed.

The life-simulation is based upon the multi-scale thermo-hygral analysis [21], where the constitutive laws are upgraded for high cycle fatigue loading and the concrete–water interaction [20,21] as shown in Figure 3. The micro-fatigue model with cyclic pore pressure was integrated into the multi-scale simulation, where the average stiffness degrades with the disintegration of the aggregates-cement composites during the fatigue simulation of concrete with condensed water. The disintegration phenomenon between aggregates and cement composites is reproduced by integrating the damage evolved with increasing local pressure between aggregates and cement paste. This local pressure, which is also computed by the constitutive model for multi-directional cracked concrete, degrades the bond between cement matrix and aggregates leading to erosion of the composite system [20].

On the basis of previous research of the freeze-thawing mechanism [23] and rate-type of fatigue modelling [24], Equation (1) [21] expresses the overall stresses of disintegrated reinforced concrete, where it is computed by integrating three constitutive models: (1) non-eroded concrete with and without cracking, (2) steel reinforcement and (3) assembly of aggregates. When there is no erosion caused by internal water impact, stresses of the element are rooted in (1) and (2). If full disintegration develops finally, the total stresses come solely from (3). Here, the erosion factor denoted by $K$ is defined to interpolate these two extreme states. Then, we have

$$\sigma_{ij} = K \cdot \sigma_{c,ij} + \sqrt{K} \cdot \sigma_{s,ij} + (1 - K) \cdot \sigma_{agg,ij}$$
$$K = e^{-Z}$$
$$dZ = -10^n \cdot (1 + f_n) \cdot p_{ampl}{}^{f_n} \cdot dp \tag{1}$$

where $\sigma_{ij}$ is total compressive stress tensor, $\sigma_{c,ij}$ and $\sigma_{s,ij}$ are stress tensors carried by cracked concrete and steel reinforcement, $\sigma_{agg,ij}$ is the stress tensor representing the fictitious aggregate particles, $Z$ is set forth as an accumulated damage of concrete in micro-pore structure, $(n, f_n)$ are coefficients related to the intersection and the slope of S–N diagrams and equal to (2.0, 0.4), respectively, $p_{ampl}$ is amplitude

of pore water pressure. The concrete and steel stresses ($\sigma_{c,ij}$ and $\sigma_{s,ij}$) are computed from the given space-averaged strain of finite element by the non-orthogonal multi-directional cracked concrete constitutive model and the aggregate stresses are computed by the multi-spring soil model [21].

During evolution of the disintegration, the cement binder is eroded and the effective stress of the concrete and steel parts is degraded until reaching zero at the same particular time of full erosion (K = 0). Here, the assembly of the remaining aggregates still can sustain compressive stresses from volumetric contraction similar to the behavior of soil particles in the geotechnical field, where the stiffness of the aggregates assembly without cement binder is estimated to be 1/100 of the normal concrete [25]. Finally, by that model, the deterioration from the elevated pressure of crack–water interaction during fatigue loading can be simulated.

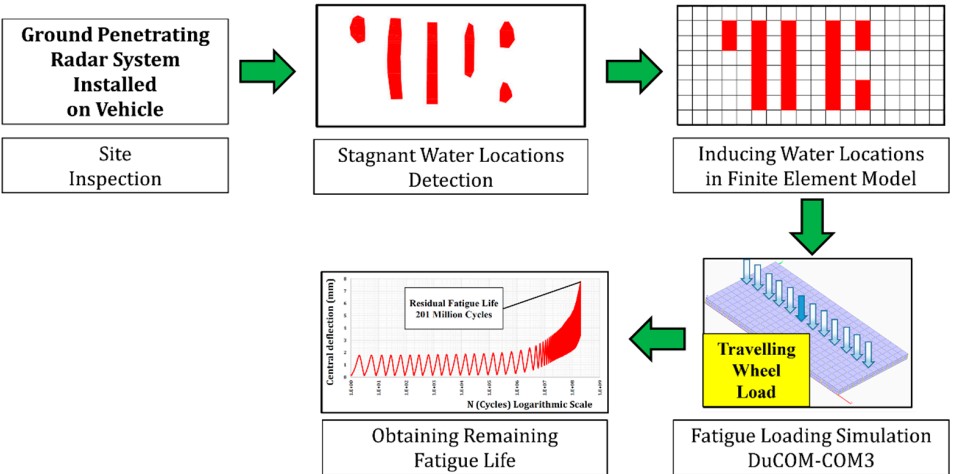

**Figure 2.** Overview of remaining fatigue life prediction methodology.

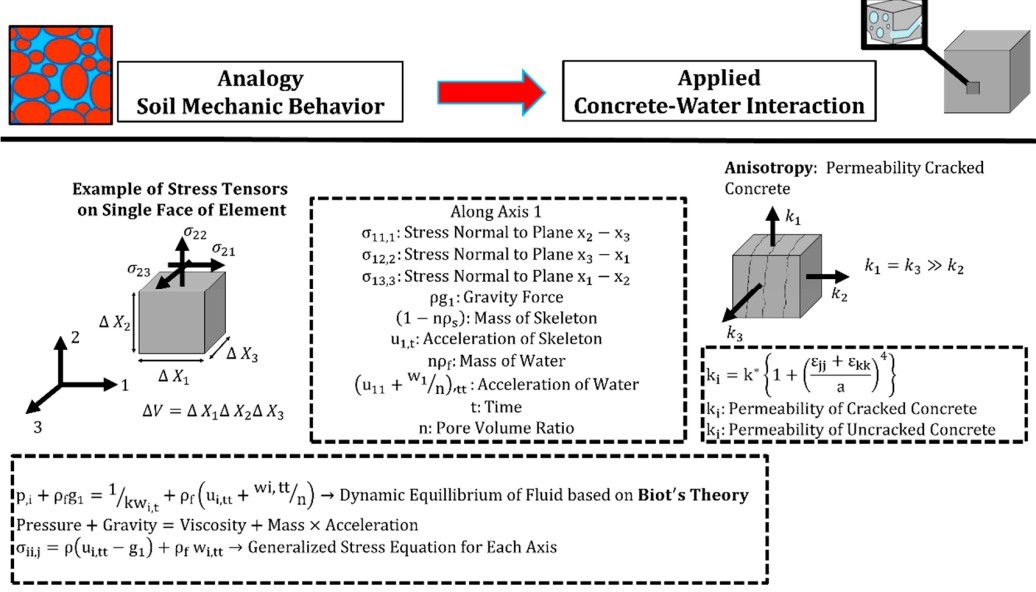

**Figure 3.** Constitutive laws of concrete–water interaction.

## 3. Specifications for the Parametric Study

### 3.1. Referential Reinforced Concrete Deck

In this study, we select a referential RC deck in reference to typical old bridge decks in Japan. Figure 4 shows the dimensions and the reinforcement arrangement of the targeted deck. A large

amount of RC decks, which are currently problematic in maintenance due to short fatigue life, were designed as one-way slab supported by side girders, while their length generally depends on several conditions such as the span of adjacent girders. In a previous research study [26], 6.0 meters' length was selected as a reference to represent a large amount of bridge infra-stocks in Japan. Thus, the authors select the same target as well.

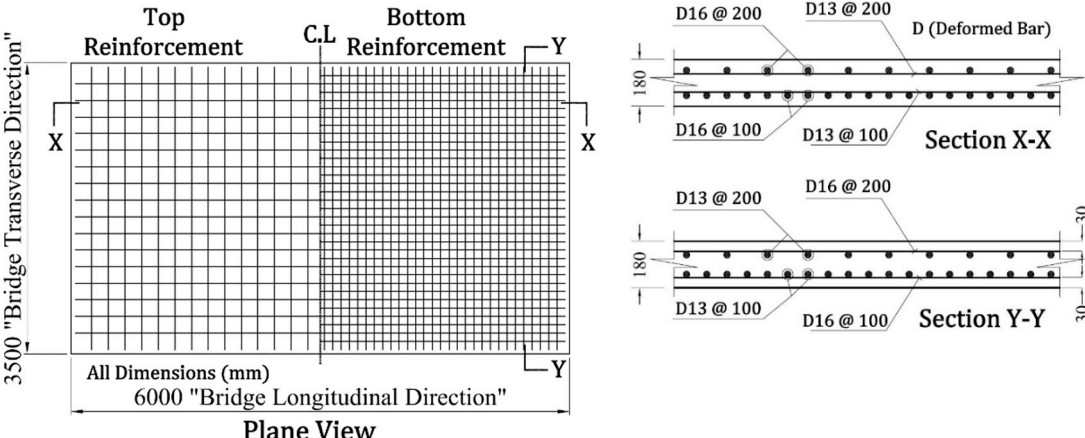

**Figure 4.** Dimensions and reinforcement of the reinforced concrete (RC) deck discussed.

### 3.2. Material Properties

Material properties of concrete and steel of the studied deck are shown in Table 1 on the basis of general design values used in the past construction practice of highway bridge decks.

**Table 1.** Material properties of the slab for analysis.

| Material Type | | Concrete | Steel Reinforcement |
|---|---|---|---|
| Young's Modulus | N/mm$^2$ | 24,750 | 205,000 |
| Compressive Strength | N/mm$^2$ | 30 | 295 |
| Tensile Strength | N/mm$^2$ | 2.2 | 295 |
| Specific Weight | kN/m$^3$ | 24 | 78 |

### 3.3. Wheel-Type Moving Load

Referring to the specification for highway bridges-Part III [27], the deck is subjected to travelling wheel-type design load of 98 kN as shown in Figure 5. Running speed of the wheel is chosen as 60 km/h, which is used to be the legal speed limit for national routes. The dimensions of the wheel are 500 and 250 mm in reference to the contact area of wheel tires.

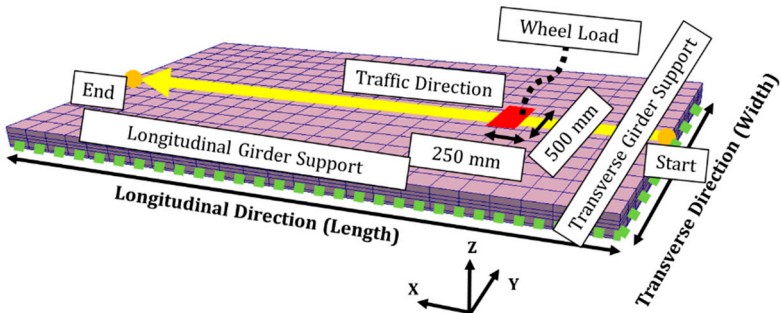

**Figure 5.** Standardized state of the RC deck discussed.

### 3.4. Failure Criterion

The fatigue limit state was specified according to the central live load deflection on the basis of past experiments and experiences [28,29]. When the live load deflection defined by Equation (2) reaches the limit state, which is equivalent to the loss of bond in flexure between concrete and steel (see Figure 6), it is judged as the fatigue failure criterion. We also accept this failure criterion in order to refer to our past research works. As this limit state live load deflection, from the experimental results, is equal to ($\cong$3) times its initial value, the authors directly apply the criterion denoted by Equation (3).

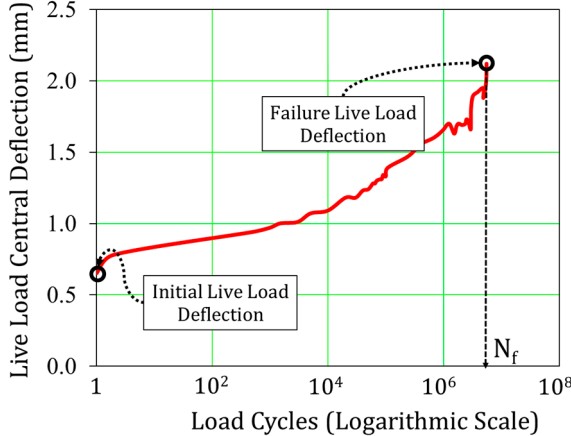

**Figure 6.** Experimentally obtained live load deflection with equivalent number of cycles.

Generally, fatigue life of RC decks is dependent on dimensions, material properties, and boundary conditions of the specimens. However, this fatigue failure criterion was empirically found to be rational since the properties of RC decks are rooted in the mere values of live load deflection. Moreover, it should be noted that the fatigue rupture of steel rebar, under moving loading, does not occur before failure of concrete [30]. Therefore, this criterion is thought to be feasible for the assessment of the failure of RC decks.

$$\delta_{L,N} = \delta_{1,N} - \delta_{2,N}, \tag{2}$$

$$\delta_{L,N} / \delta_{L,0} \geq 3.0, \tag{3}$$

where $\delta_{L,N}$ is central live load deflection at the $N^{th}$ of cycles, $\delta_{1,N}$ is central total deflection at the $N^{th}$ of cycles at the loading step, $\delta_{2,N}$ is central total deflection at $N^{th}$ of cycles at unloading step, $\delta_{L,0}$ is initial live load deflection, and $N_f$ is the failure number of cycles corresponding to ($\delta_{L,N}$ from Equation (3)).

### 3.5. Numerical Model

The simulation model is discretized with finite elements by using the open code program "FABriS" [26], which is specialized for the wheel-type moving load on RC bridge decks. Mesh size is chosen to be 250 × 250 mm in the *x–y* plane, and the number of layers in the *z*-direction is four as shown previously in Figure 5. The slab is supported by hinges at the boundaries of the RC deck. The studied cases of the effect of wetting locations on the fatigue life of the RC deck are based on varying the wetting locations among the top layer of the RC deck, as shown in Figure 7, since the radar system technique can efficiently obtain this kind of information from site [22]. The simulation explained as above has been validated by experiments and site inspection data [17].

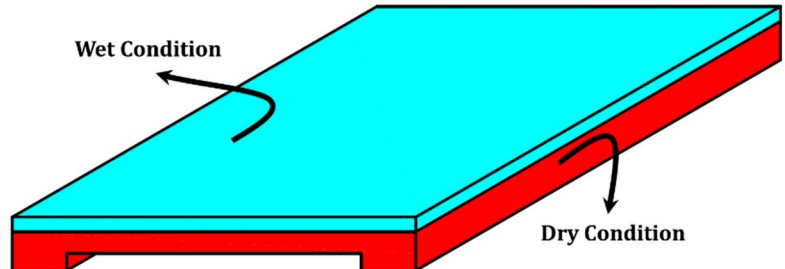

**Figure 7.** Standardized states for stagnant water studied cases.

## 4. Dry and Wet Ambient Conditions

The fatigue life of the referential RC deck in dry conditions and fully submerged was investigated in previous research [17,26,31]. However, this study targets the effect of wetting locations only at the upper entire surface layer of the RC deck. Figure 8 shows the relation of load cycles and the central live load deflection of three cases: dry, upper mesh layer "wet", and full submerged conditions, where their remaining fatigue lives are 221.49, 8.16, and 2.93 million cycles, respectively, as shown in Table 2.

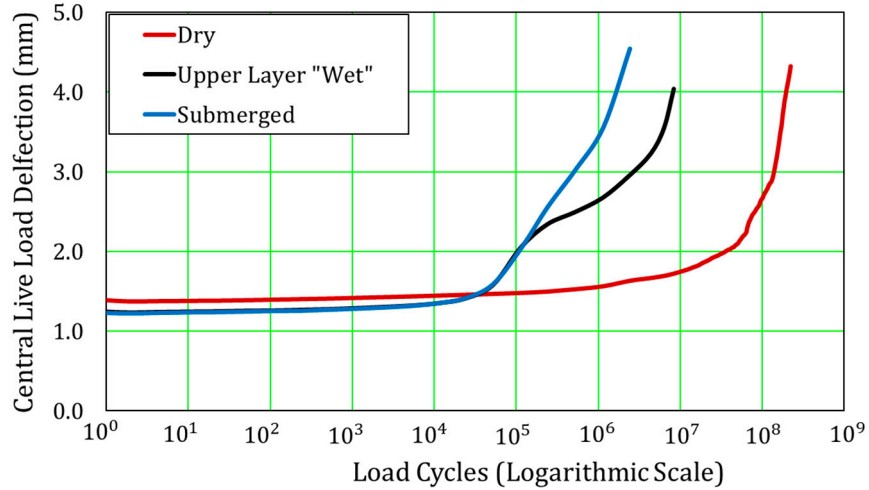

**Figure 8.** Load cycles and central live load deflection of dry and wet ambient conditions.

**Table 2.** Remaining fatigue life of dry and wet ambient conditions.

| Case | Remaining Fatigue Life (Million Cycles) | Reduction in Life (Compared to Dry Case) |
|---|---|---|
| Dry Condition | 221.49 | 1.0 |
| Upper Layer (Wet) | 8.16 | 1/27 |
| Submerged | 2.93 | 1/75 |

The pore-mechanics based simulation can indicate the profile of erosion by K factor (Equation (1)) at the central zone of the referential RC deck (upper layer "wet" in Table 2). Concrete erosion is severe around the path of the moving wheel as shown in Figure 9, and the concrete is fully eroded at 70,000 cycles as shown in Figure 10. The computed erosion profile matches the reality of the RC decks in the laboratory. Figure 11 shows the rise and decay of the pore water pressure. When the concrete composite is more or less sound as solid, pore pressure may rise according to the external load. However, when the erosion evolves much, the pore water pressure cannot rise because the water may easily pass through an assembly of cracks. In accordance with the erosion, we have varying compressive stresses of concrete at the top layer central elements and tensile stresses of reinforcement at the bottom ones as shown in Figures 12 and 13, respectively.

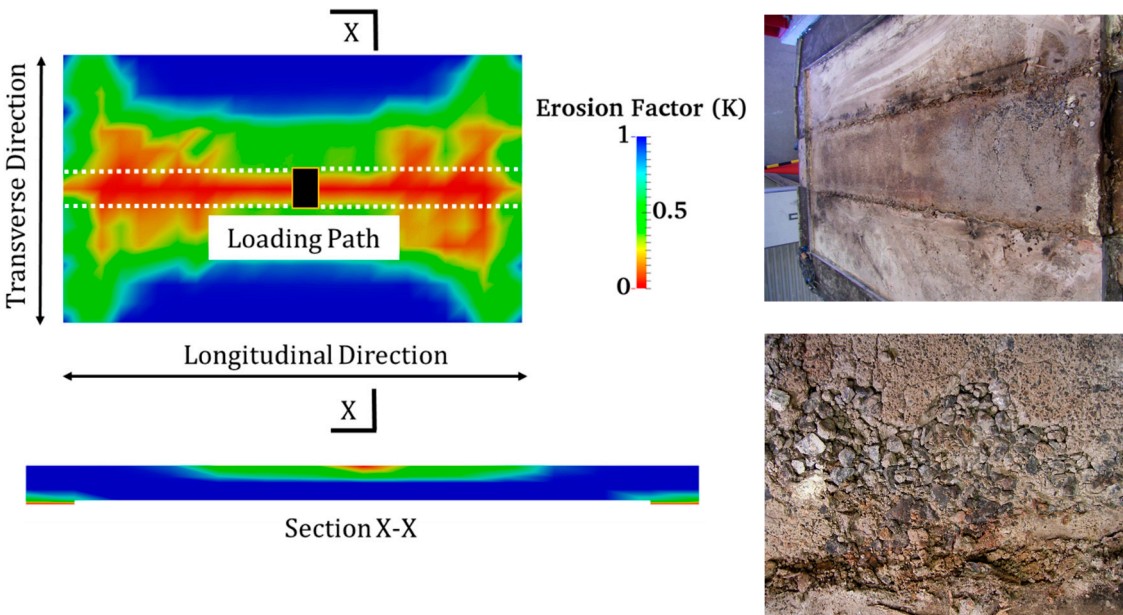

**Figure 9.** Multi-scale simulation of RC deck under fatigue loading in upper layer "wet" case.

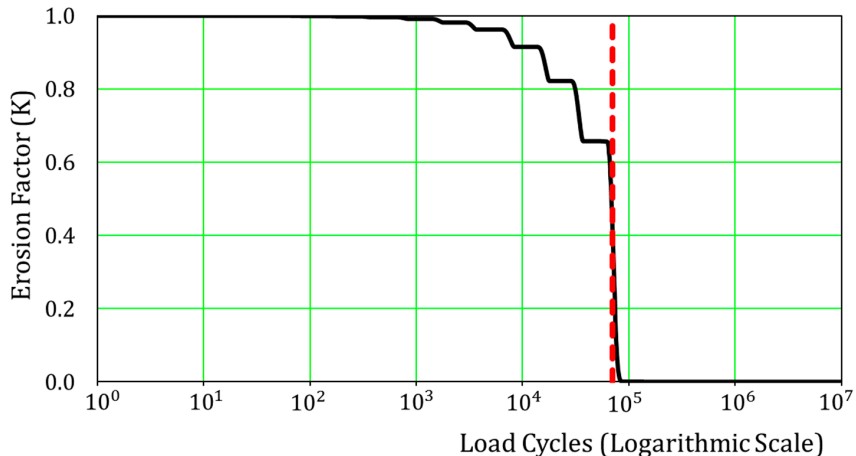

**Figure 10.** Load cycles and the erosion factor of the central zone of the RC deck.

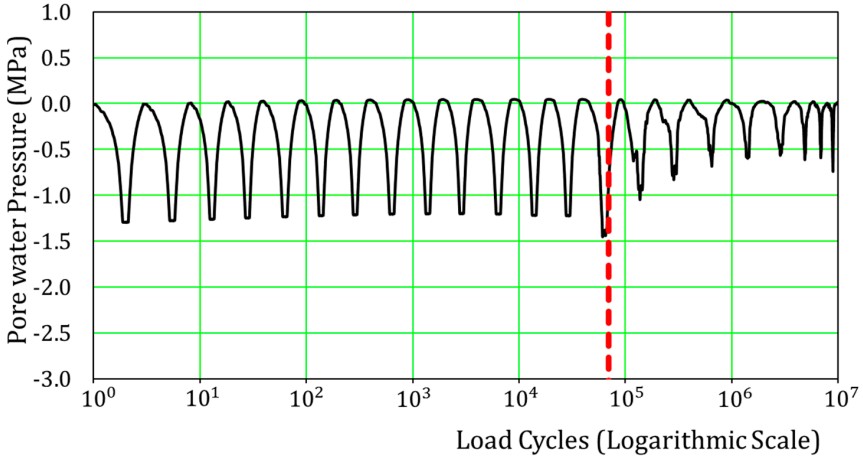

**Figure 11.** Load cycles and the pore water pressure of the central zone of the RC deck.

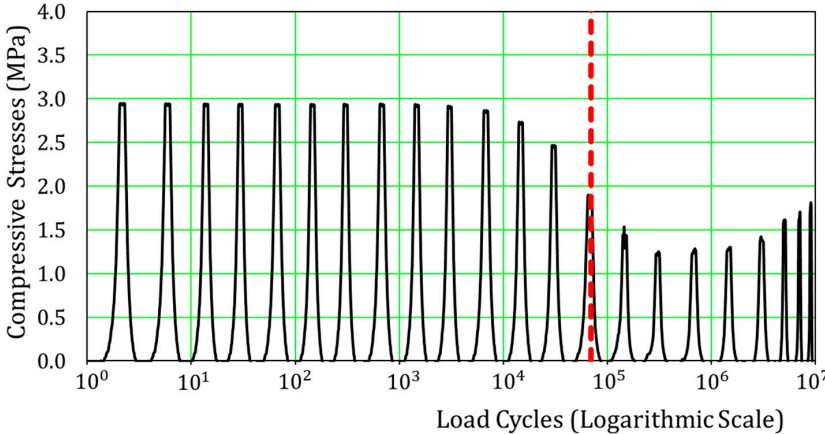

**Figure 12.** Load cycles and the top compressive stresses of the transverse direction of the central zone of the RC deck.

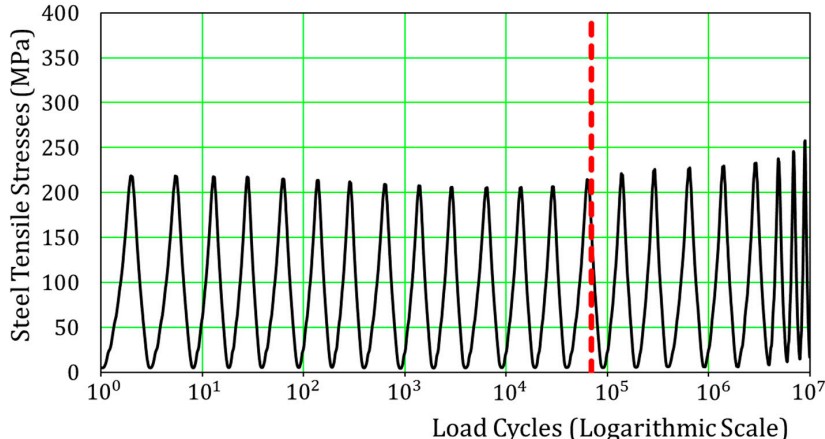

**Figure 13.** Load cycles and the bottom-steel tensile stresses of the central zone of the RC deck.

## 5. Case Study

Figure 14 shows the non-uniform patterns of the locations of stagnant water (30 cases) at the top surface of studied RC deck. The relation of the wetting rate at the top surface (WR) expressed in Equation (4), and the computed fatigue lives is shown in Figure 15, where their detailed remaining fatigue lives, normalized fatigue life by the fully dry condition, and WR indices are listed in Table 3. The simulation results prove the wide range of fatigue life despite of the same WR aiming at the significance of wetting patterns on the fatigue life. It is clear that the fatigue life is significantly reduced when the wetting locations are at the central zone of the wheel loading path as the following cases: (3), (13), (16), (20), (27), and (29).

On the other hand, the impact of stagnant water is significantly reduced, when the wetting locations are near the sides of the deck, away from the wheel loading path, as the following cases: (4–9), (17–19), and (26). It is demonstrated that the negative impact of the stagnant water on the fatigue life is reduced as the wetting locations start to be farther from the central zone of the wheel-loading path, as shown in Figure 16, where X is the longitudinal distance of the central wetting location to the center of the RC deck.

$$WR\% = \frac{\sum_{k=1}^{k=n}(E_k)}{n} \times 100, \tag{4}$$

where WR% is the wetting rate among the top mesh layer of the RC deck, k is the $k^{th}$ element at the top surface, $E_k$ is environmental condition of the $k^{th}$ element (0 for dry, 1 for wet), and n is the total number of elements at the top surface of the deck (336 is this study).

For further investigation for the wetting pattern, a comparison has been made between case (5) that has low water impact and case (16) that has high impact, where the difference in their fatigue lives is around 1.6 times despite of the quite close value of WR. Figure 17 shows the development of the central live load deflection with loading cycles for cases (5) and (16), where their remaining fatigue lives are 160.0 and 99.8 million cycles, respectively. By investigating the rate of erosion of the concrete at the wetting locations for both cases (see Figure 18), it is obvious that the erosion starts earlier in case (16) than case (5). This is also clear in the rise of the pore water pressure for each case at the same particular time of reaching full erosion (K = 0), as shown in Figure 19. This can be explained by checking the average principal strains at the bottom surface of the RC deck just below their wetting locations, as shown in Figure 20.

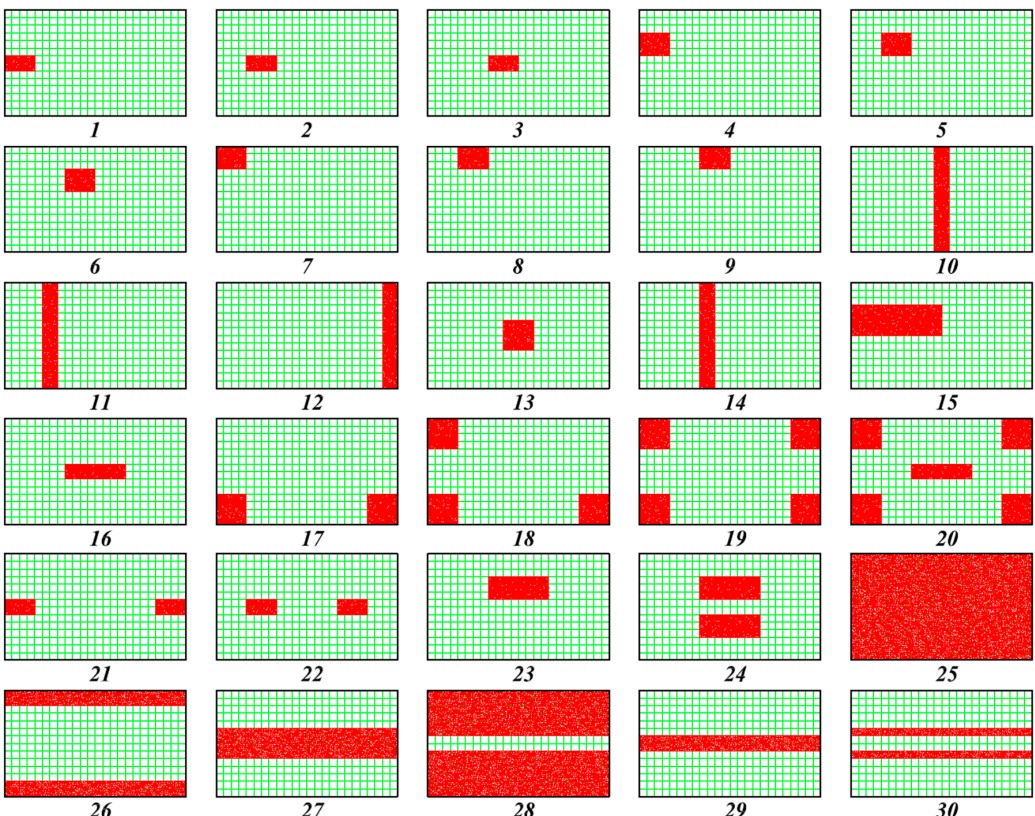

**Figure 14.** Location patterns of stagnant water.

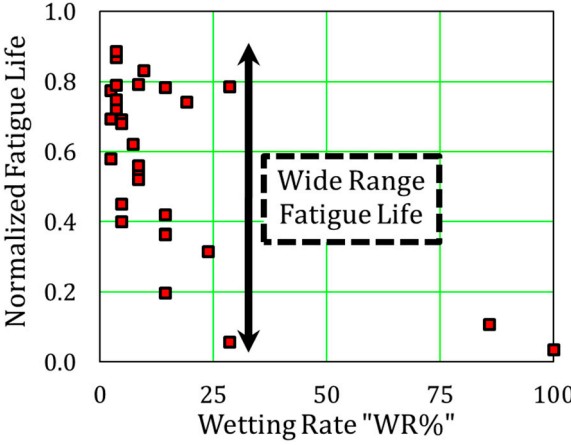

**Figure 15.** Relation of wetting rate at the top layer of RC deck and the fatigue life.

**Table 3.** Studied remaining fatigue lives and wetting rates.

| Case | Life "Million Cycles" | Normalized "Dry" | WR% | Case | Life "Million Cycles" | Normalized "Dry" | WR% |
|---|---|---|---|---|---|---|---|
| 1 | 171.7 | 0.78 | 2.4 | 16 | 99.8 | 0.45 | 4.8 |
| 2 | 153.8 | 0.69 | 2.4 | 17 | 184.0 | 0.83 | 9.5 |
| 3 | 128.3 | 0.58 | 2.4 | 18 | 173.5 | 0.78 | 14.3 |
| 4 | 174.9 | 0.79 | 3.6 | 19 | 164.5 | 0.74 | 19.0 |
| 5 | 160.0 | 0.72 | 3.6 | 20 | 69.7 | 0.31 | 23.8 |
| 6 | 165.7 | 0.75 | 3.6 | 21 | 153.1 | 0.69 | 4.8 |
| 7 | 193.6 | 0.87 | 3.6 | 22 | 150.6 | 0.68 | 4.8 |
| 8 | 192.6 | 0.87 | 3.6 | 23 | 137.6 | 0.62 | 7.1 |
| 9 | 196.5 | 0.89 | 3.6 | 24 | 81.0 | 0.37 | 14.3 |
| 10 | 115.5 | 0.52 | 8.3 | 25 | 8.2 | 0.04 | 100.0 |
| 11 | 121.7 | 0.55 | 8.3 | 26 | 174.0 | 0.79 | 28.6 |
| 12 | 175.7 | 0.79 | 8.3 | 27 | 13.1 | 0.06 | 28.6 |
| 13 | 88.8 | 0.40 | 4.8 | 28 | 24.2 | 0.11 | 85.7 |
| 14 | 124.2 | 0.56 | 8.3 | 29 | 43.6 | 0.20 | 14.3 |
| 15 | 93.1 | 0.42 | 14.3 | 30 | 80.6 | 0.36 | 14.3 |

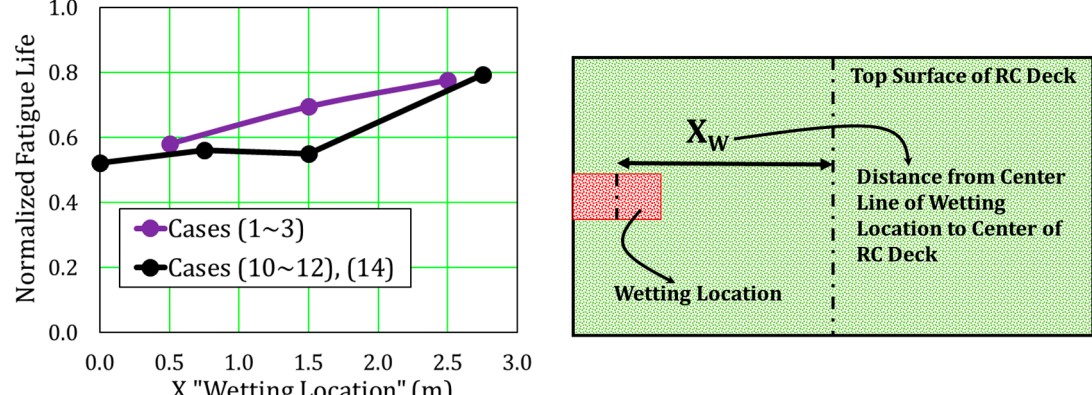

**Figure 16.** Effect of wetting locations on the fatigue life of RC decks.

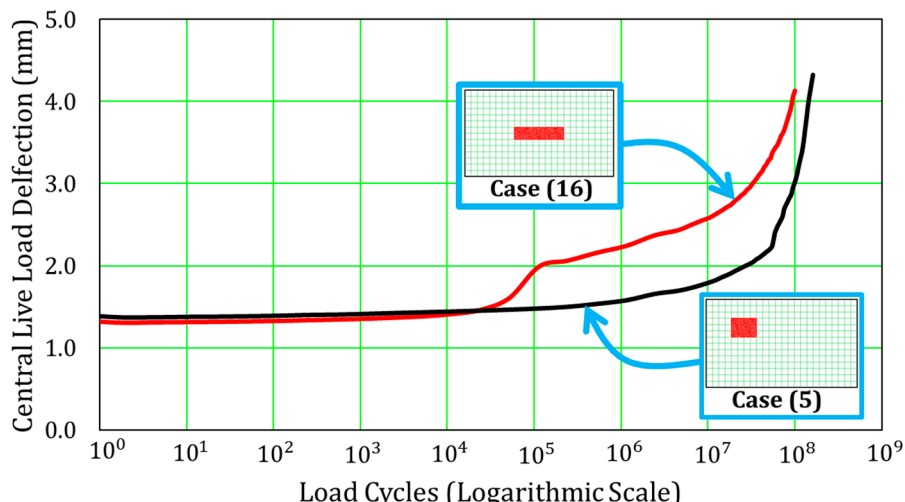

**Figure 17.** Load cycles and the central live load deflection of cases (5) and (16).

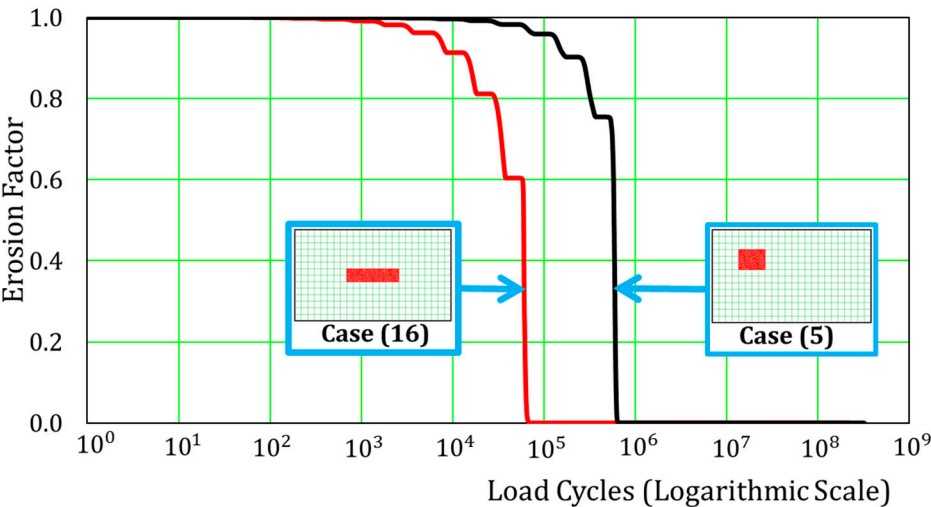

**Figure 18.** Load cycles and the erosion factor of the upper zone of cases (5) and (16) at their wetting locations.

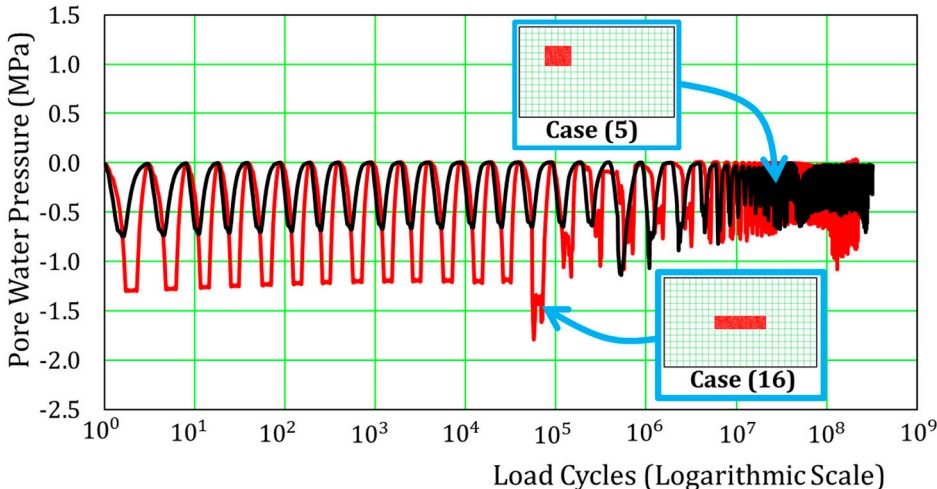

**Figure 19.** Load cycles and the pore water pressure of the upper zone of cases (5) and (16) at their wetting locations.

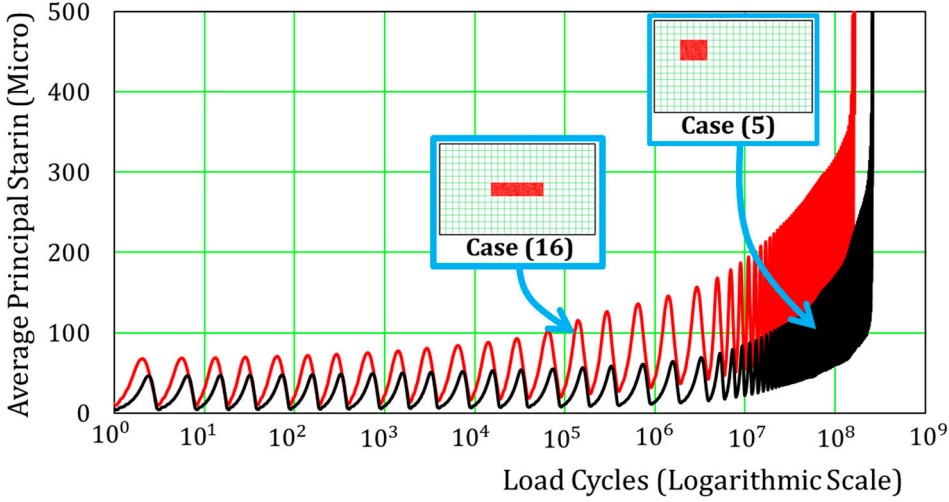

**Figure 20.** Load cycles and the average principal strains of the bottom zone of cases (5) and (16) at their wetting locations.

It is found that the amplitude of the principal strains for case (16) is higher than that of case (5), since it is the location of maximum bending moment, where the flexure cracks occur with great strain. On the other hand, flexural cracks are dispersing as we go farther from the central zone of the RC deck, where the reduction of principal strains for case (5) is clearly seen. The segregation of aggregates and cement paste is dependent on the opening and closure of the cracks, where the pore water pressure increases at the closure of the cracks leading to the damage of the local interface between the aggregates and the cement binder, as shown in Figure 21. Thus, more flexure cracks and deformation amplitudes of case (16) lead to the negative impacts on the remaining fatigue life compared to case (5).

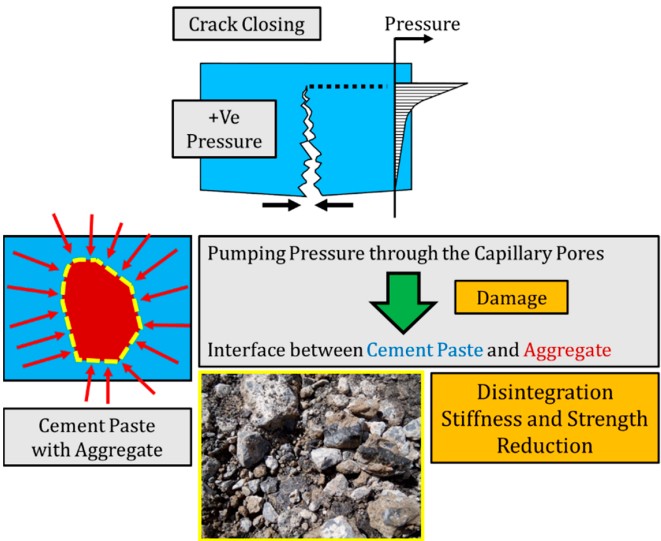

**Figure 21.** Mechanisms of disintegration of concrete under high cycle fatigue.

## 6. Hazard Map for Engineering Practice

In this section, the authors try to build a hazard map of water for the wetting locations based on the simulation results. Cases (1~9), besides the laws of symmetry, are utilized to build the hazard map since these cases cover one quarter of the top surface of the RC deck without overlapping, as shown in Figure 22. The normalized fatigue life compared to the full dry state of the reference case is normalized between 0 and 1 to achieve 2D contour lines of the hazard map with the upper bound of (1) and the lower one of (0), which denote the lowest and the highest hazard regions, respectively, as shown in Table 4. Finally, a hazard map for the wetting locations is introduced, as shown in Figure 23. It is demonstrated that the central zone of the wheel-loading path is highly hazardous. The introduced hazard map allows the bridges' inspectors to evaluate the risk of the RC decks by utilizing the visual and/or non-destructive inspection [22,32].

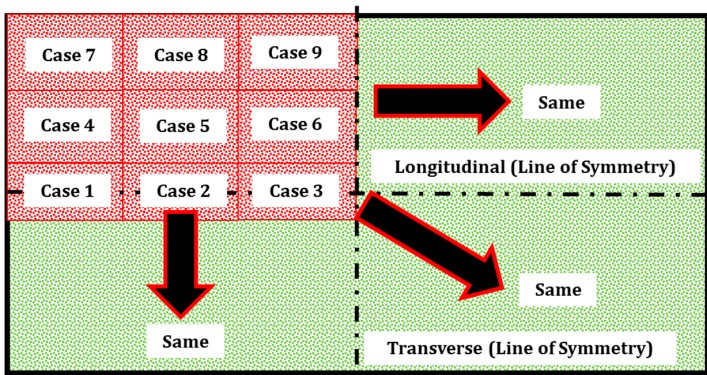

**Figure 22.** Methodology for achieving the hazard map.

**Table 4.** Utilized cases for building the hazard map.

| Case | Life "Million Cycles" | Normalized "Dry" | Normalization (0 and 1) |
|------|------------------------|-------------------|--------------------------|
| 1 | 171.7 | 0.78 | 0.65 |
| 2 | 153.8 | 0.69 | 0.35 |
| 3 | 128.3 | 0.58 | 0.00 |
| 4 | 174.9 | 0.79 | 0.68 |
| 5 | 160.0 | 0.72 | 0.45 |
| 6 | 165.7 | 0.75 | 0.55 |
| 7 | 193.6 | 0.87 | 0.94 |
| 8 | 192.6 | 0.87 | 0.94 |
| 9 | 196.5 | 0.89 | 1.00 |

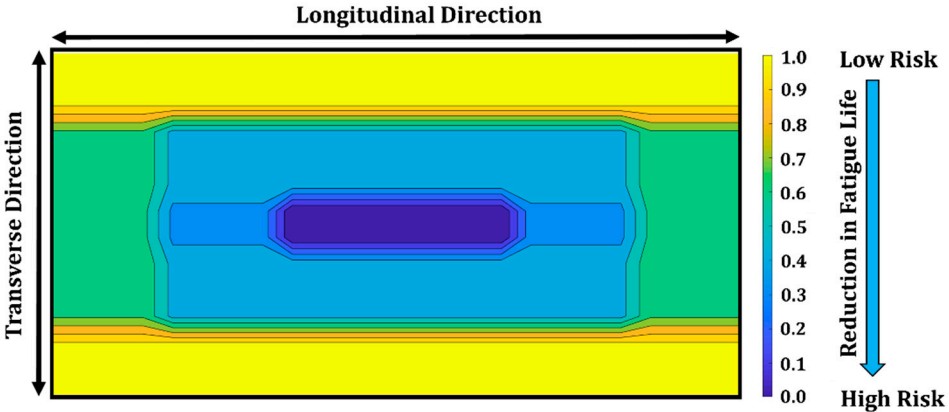

**Figure 23.** Hazard map for the wetting locations of higher risk.

## 7. Predictive Correlation

On the basis of the proposed hazard map, a predictive correlation for fatigue life of RC decks, based upon site inspected wetting locations, is introduced. The top surface of the RC deck is divided into four regions (see Figure 24), where WR index (Equation (4)) is calculated for each zone. Then, a predictive damage index (D.I.) is introduced by integrating the WR index for the four regions, where each region is assigned to different weight based on its risk impact, as shown in Equation (5). Finally, nonlinear correlation (Equation (6)) is introduced for the remaining fatigue life as a function of the proposed damage index. Figure 25 shows the relation of the damage index and the fatigue life of the studied cases, where the regression coefficient, coefficient of variation (C.O.V), and prediction interval of variance of 95% (P.I.) of the proposed correlation are 0.91, 21%, and 17%, respectively. The proposed correlation offers to analyze the remaining fatigue life of RC deck based on site inspected wetting locations with high accuracy:

$$\text{D.I.} = 0.55 \cdot WR_1 + 0.23 \cdot WR_2 + 0.17 \cdot WR_3 + 0.05 \cdot WR_4, \tag{5}$$

$$\text{Life} = 1.49 + 1.43 \cdot \tanh(-0.36 - 0.24 \cdot \text{D.I.}), \tag{6}$$

where D.I. is the damage index of water impact, and ($WR_1$, $WR_2$, $WR_3$, $WR_4$) are wetting rates for the regions shown in Figure 24 and calculated from Equation (4).

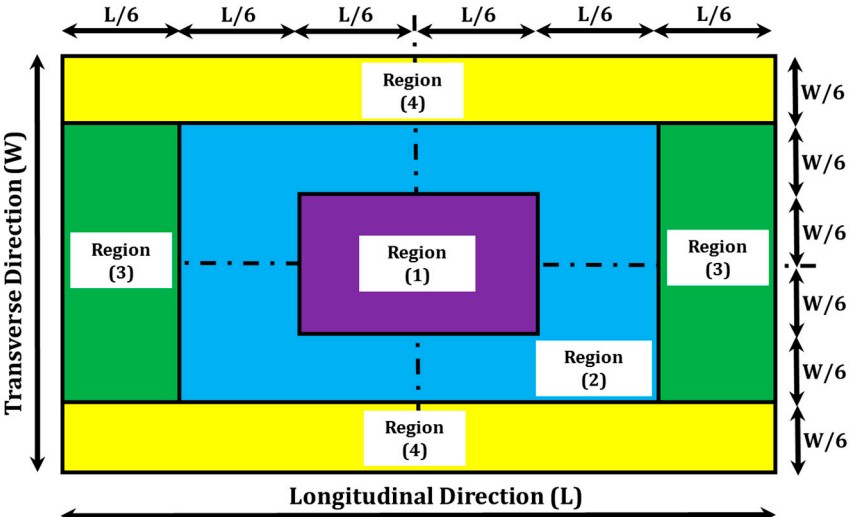

**Figure 24.** Damage index concept.

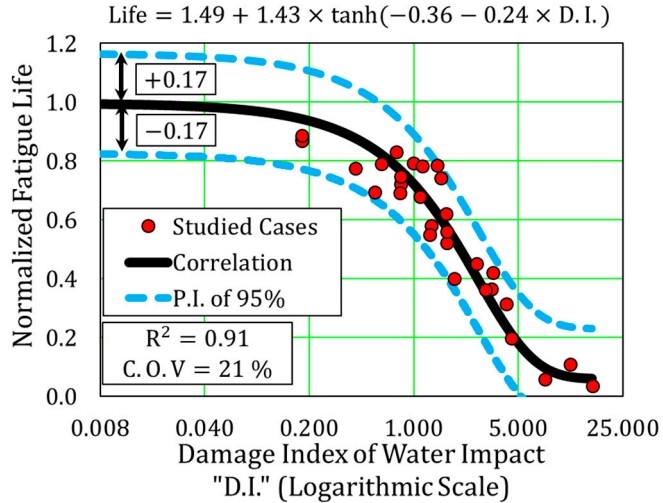

**Figure 25.** Relation of the damage index of water impact and the normalized fatigue life.

## 8. Conclusions

The impact of wetting by the stagnant water on the remaining fatigue life of the RC decks is studied by utilizing the multi-scale simulation, which was previously validated by experiments and site experiences, and the following conclusions are earned.

1.  The patterns of wet and dry areas have a great influence on the remaining fatigue life of the RC decks.
2.  It is found that the central zone of the wheel-loading path is the wetting location of higher risk on the fatigue life.
3.  The negative impacts of stagnant water reduce gradually as the wetting locations go farther from the central zone of the wheel-loading path, where these impacts tend to significantly reduce at the sides of the RC deck away from the wheel-loading path.
4.  A hazard map for the wetting locations of higher risk is proposed based on the simulation results, which is beneficial for bridges' inspectors.
5.  A predictive correlation is proposed for fatigue life prediction of RC deck based on site inspected wetting locations with high accuracy, which fulfill the engineers' needs to conduct the risk assessment of RC decks during maintenance.

It should be noted that the scope of the proposed predictive correlation is to achieve the risk magnitude of inspected wetting locations with regard to the reference RC deck. For other RC decks, whose dimensions and material properties differ from the referential RC deck, the sensitivity of the proposed correlation will be checked in the near future regarding its generalization to deal with any RC deck. Then, as a next future step, the authors aim to couple the introduced predictive model in this study with our previous proposed artificial intelligence (AI) models that deal with fatigue life prediction based on site-inspected cracks [17,26,31].

**Author Contributions:** E.F. conducted and analyzed the numerical studied cases; Y.T and K.M. supervised over the analytical process and developed the simulation program; E.F. and K.M. wrote the paper.

**Funding:** This study was financially supported by the Council for Science, Technology, and Innovation, "Cross-ministerial Strategic Innovation Promotion Program (SIP), Infrastructure Maintenance, Renovation, and Management" granted by the Japan Science and Technology Agency (JST).

**Acknowledgments:** The authors extend their appreciation to Yozo Fujino of Yokohama National University and the University of Tokyo, for his valuable advice and encouragement to bridge the multi-disciplines in engineering. Many thanks to Takahiro Yamaguchi of the University of Tokyo for his great discussions and valuable comments about ground penetrating radar system technology.

**Conflicts of Interest:** The authors declare no conflict of interest.

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
