# Peer review of "Fatigue Life of RC Bridge Decks Affected by Non-Uniformly Dispersed Stagnant Water"

_applsci, doi:10.3390/app9030607_

Round 1

Reviewer 1 Report

The paper concerns a new approach for evaluating the reduction of the residual fatigue life of RC bridge deck due to non-uniform distribution of stagnant waters. With reference to a case study, starting from the location of wetting areas obtained through GPR in-situ inspections, and by a multi-scale simulation, the Authors obtain an hazard map for the wetting locations of higher risk and a predictive fatigue life correlation.

I have found the subject of the paper is very interesting, and the proposed approach appears to be promising for further generalizations, aimed at supporting practical risk assessment of RC bridge decks. Anyway, in my opinion the paper may strongly benefit by the following revisions:

1) give a broader exposition of the mechanical significance of Equation 1 (by the way, I have not found Equation 1 in reference [19], as recalled in line 78);

2) give a broader exposition of the failure criterion adopted in Sect. 3.4;

3) the title of reference [21] is wrong.

Author Response

Dear Reviewer 1,

    Thank you very much for your comments. Please find the attached response. Thank you!

Reviewer 2 Report

The paper studies fatigue life of reinforced concrete bridge decks affected by non-uniformly dispersed stagnant water. The authors propose a methodology to systematically compute fatigue life for various locations of stagnant water at upper layer of RC decks. The structure of the paper is appropriate. The introduction and English language, style, and formal appearance primarily prevent the reviewer from recommending paper acceptance.

1. Introduction
The literature review part (or background, respectively) must considerably be extended. The description of work done in the field is restricted to 10 sentences. Almost two thirds of the references are self-citations of the authors, precisely 7, 8, 9, 10, 14, 15, 16, 17, 18, 19, 20, 21, 24, 25, 26, 27, 28, and 29. In particular, the reviewer does not see a reason why to cite 8, 9, 10, 14, 15, 16, 17. Here, citing one study of the authors should be sufficient.

2. English language, style, and formal appearance
Deficiencies in English writing and formal appearance make the paper sometimes hard to read. It should be thoroughly revised in these directions.

2.1 In the paper, “the” definite article is overused causing ambiguities. The definite article, in case of countable nouns, should only be used if the entity has already been introduced or if the authors refer to a distinct entity; if referring to entities in a generic manner, plural should be used. This issue should be solved throughout the paper (e.g. “Abstract: The stagnant water on reinforced concrete (RC) decks reduces...” --> “Abstract: Stagnant water on reinforced concrete (RC) decks reduces...”
2.2 The same font size and font typeface should be used in the labels/callouts of every figure.
2.3 The authors are invited to double-check the use of prepositions (e.g. “Signal processing for the collected data...” --> “Signal processing for the collected data...”).
2.4 Capitalization of figure callouts and labels should be consistent (e.g. “Upper layer” --> “Upper Layer”).
2.5 Two references, such as [14-15], should be separated by comma [14, 15].
2.6 The possessive case sounds sometimes awkward and its overuse should be avoided in academic writing (“concrete’s performance”, “water’s location”, “cracks’ surface”, “deck’s fatigue life”, “concrete’s strength”, “wheel tires’ contact area” etc.).

Author Response

Dear Reviewer 2,

    Thank you very much for your comments. Please find the attached response. Thank you!

Reviewer 3 Report

Dear Authors, 

   I am pleased to read your paper and share my opinion regarding the first manuscript. The paper, with a few English writing mistakes, is well-written and the contents are interesting for bridge engineers. The methodology and the results are clearly presented with sufficient details, as well. I would recommend the paper for publication in the present form after considering the other reviewers' comment. 

Some suggestions for English writing improvement:

- Line 026: 'waterproofing' is a single word.

- Line 037: use a/the before "wet".

- Line 044, 137: use a/the before"site".

- Line 066: 'later section' is correct.

- Line 071: 'integrated into' is recommended. 

- Line 104: 'a large amount of' is recommended.

- Line 126: 'an equivalent..' is recommended. 

- Line 132: add a/the before "wheel".

- Line 145: 'fully submerged' is correct. 

- Line 155: add a/the before "laboratory", and "concrete".

- Line 180: add a/the before "impact".

- Line 201: replace "far more" with 'farther'.

- Line 253: 'offers to analyze' is correct, not "offers analyzing".

- Line 267/271/275: use a/the before "central", "wheel-loading", and "risk".

Regards,

Author Response

Dear Reviewer 3,

    Thank you very much for your comments. Please find the attached response. Thank you!

Round 2

Reviewer 2 Report

In the first review, it has been recommended to “considerably extend” the review part (or background, respectively). In the response to this review, the authors have added another self-citation, describing their own work, to the background section and to further sections, instead of providing sufficient background. Also, the authors still prefer to cite their own work in 50% (14/28) of the references, instead of including relevant references of work conducted in the field.

Author Response

In the first review, it has been recommended to “considerably extend” the review part (or background, respectively). In the response to this review, the authors have added another self-citation, describing their own work, to the background section and to further sections, instead of providing sufficient background. Also, the authors still prefer to cite their own work in 50% (14/28) of the references, instead of including relevant references of work conducted in the field.

 AUTHORS' ANSWER

For the review of engineering problems on bridge decks in the world, the background of the deteriorations of bridge slabs was further updated in the first paragraph of introductory section, and we try to clarify the problem of bridge decks especially constructed in seismic hazardous regions as follows (Yellow part as below is embedded in the manuscript) .

Reinforced concrete (RC) bridge decks suffer from high deteriorations due to environmental attacks besides traffic loading, where corrosion, freeze & thawing, alkali silica reaction, and shrinkage & thermal cracking were reported to be significant on the reduction of life of RC decks [1-9].

On the other hand, in high seismic risk countries like Japan, thickness of RC decks were aimed to be thinner in order to reduce the inertia forces at earthquakes to satisfy earthquake resistant design requirements since bridge slabs are main source of seismic loads to bridges. These limited-thicknesses of decks (less than 200 mm) were constructed in 1960-70s, where enormous numbers of highway bridges were built. After around half century, degradation of bridge decks has been observed from accumulated loads of daily traffics, where these deteriorations may reduce the safety of users.

Previous researches report that RC slabs exposed to moving loads are extremely deteriorated compared to those exposed to fixed-point pulsating ones. The reversal cyclic-shear along crack planes induced by moving wheel type loading rapidly deteriorates the shear transfer of aggregates interlock along concrete cracks [10]. Finally, RC slabs speedily lose their stiffness until total failure. Thus, we have common issues in view of bridge deterioration where mechanical fatigue loads and environmental actions develop together with more or less interaction.

It should be noted that the performance of RC decks can be upgraded by utilizing pre-stressing techniques [11] for newly constructed bridges. In fact, traffic-induced cracking is suppressed and water may not come inside crack gaps of concrete. However, we have to face serious problems as such that damaged RC decks of many bridges cannot be easily replaced since it will directly disturb the traffic flow leading to social problems. Therefore, old RC decks shall be retrofitted and/or limitedly replaced for extending the lifetime of RC decks on the basis of reliable maintenance plans.

References outside our research group are added (total 11) as follows. On the other hand, references numbers (10, 13, 14, 24, 26, 27) of the previously submitted draft “self-citations” are omitted.

1.           Val, D. V.; Stewart, M. G.; Melchers, R. E. Effect of reinforcement corrosion on reliability of highway bridges. Engineering Structures 1998, 20(11), 1010-1019.

2.           Vassie, P. Reinforcement corrosion and the durability of concrete bridges. In Proceedings of Institution of Civil Engineers, Pt 1, 76, 1984.

3.           Cady, P. D.; Weyers, R. E. Chloride penetration and the deterioration of concrete bridge decks. Cement, concrete and aggregates 1983, 5(2), 81-87.

4.           Freyermuth, C. L.; Klieger, P.; Stark, D. C.; Wenke, H. N. Durability of concrete bridge decks-a review of cooperative studies. Highway Research Record 1970, 328.

5.           Lindquist, W. D.; Darwin, D.; Browning, J.; Miller, G. G. Effect of cracking on chloride content in concrete bridge decks. American Concrete Institute 2006.

6.           Shah, S. P.; Weiss, W. J.; Yang, W. Shrinkage Cracking--Can It Be Prevented?. Concrete International 1998, 20(4), 51-55.

7.           Sakulich, A. R.; Bentz, D. P. Increasing the service life of bridge decks by incorporating phase-change materials to reduce freeze-thaw cycles. Journal of Materials in Civil Engineering 2011, 24(8), 1034-1042.

8.           Detwiler, R. J.; Kojundic, T.; Fidjestol, P. Evaluation of bridge deck overlays. CONCRETE INTERNATIONAL-DETROIT- 1997, 19, 43-46.

9.           Ulm, F. J.; Coussy, O.; Kefei, L.; Larive, C. Thermo-chemo-mechanics of ASR expansion in concrete structures. Journal of engineering mechanics 2000, 126(3), 233-242.

10.          Perdikaris P. C; Beim S. RC Bridge Decks Under Pulsating and Moving Load. ASCE-Journal of Structural Engineering 1988, 114.

11.          Poston, R. W.; Carrasquillo, R. L.; Breen, J. E. Durability of post-tensioned bridge decks. Materials Journal 1987, 84(4), 315-326.